# Unified Model for Code-Switching Speech Recognition and Language Identification Based on Concatenated Tokenizer

**Kunal Dhawan, Dima Rekesh, Boris Ginsburg**
NVIDIA, Santa Clara, USA
{kdhawan, drekesh, bginsburg}@nvidia.com

## Abstract

Code-Switching (CS) multilingual Automatic Speech Recognition (ASR) models can transcribe speech containing two or more alternating languages during a conversation. This paper proposes (1) a new method for creating code-switching ASR datasets from purely monolingual data sources, and (2) a novel Concatenated Tokenizer that enables ASR models to generate language ID for each emitted text token while reusing existing monolingual tokenizers. The efficacy of these approaches for building CS ASR models is demonstrated for two language pairs, English-Hindi and English-Spanish, where we achieve new state-of-the-art results on the Miami Bangor CS evaluation corpus. In addition to competitive ASR performance, the proposed Concatenated Tokenizer models are highly effective for spoken language identification, achieving 98%+ accuracy on the out-of-distribution FLEURS dataset.

## 1 Introduction

Automatic Speech Recognition (ASR) systems are moving from specialized monolingual models to ASR architectures capable of handling multiple languages simultaneously (Weng et al., 1997; Waibel et al., 2000; Kannan et al., 2019; Li et al., 2022; Pratap et al., 2023). Code-Switching (CS) is a special category of multilingual speech in which two or more languages or varieties of languages are used in the same utterance. It can further be divided into two categories: *inter-sentential* code-switching where the switching between languages happens predominantly at the sentence boundaries and *intra-sentential* code-switching, which happens within the sentence (Myers-Scotton, 1989).

Most of the work in code-switching ASR is dependent on the availability of a good quality code-switching speech corpus (Sitaram et al., 2019). One of the questions that we explore in this paper is: how to better utilize the readily available monolingual speech corpora and train CS ASR systems that can perform well in real-world code switching scenarios.

Text post-processing after ASR, e.g. punctuation and capitalization (Guerreiro et al., 2021) and inverse text normalization (Sunkara et al., 2021), is another important problem for multilingual and CS speech systems. Such post-processing is harder than the monolingual scenario as it requires accurate language identification in addition to transcript generation. Traditionally, separate Language Identification (LID) and ASR models have been trained for the task, usually with a common acoustic encoder. Li et al. (Li et al., 2019) was one of the first few works to propose an end-to-end architecture for intra-sentential CS ASR. They trained two separate monolingual ASR systems and a frame-level LID model. The posteriors of ASR models were adjusted with the LID scores and greedy decoding was used without any language model rescoring. In (Ali et al., 2021), the authors proposed to use multigraph weighted finite state transducers, which was shown to be more effective than Transformer-based systems for the intra-sentential CS. Recent works (Seki et al., 2019), (Radford et al., 2022) approach this problem differently by introducing special LID symbols such as [EN] [ES], that are added to the vocabulary for language identification. These symbols are predicted either at the start of the utterance to identify which language the decoded text belongs to (Radford et al., 2022), or during the utterance to mark spans of decoded tokens belonging to each language (Seki et al., 2019).

In this paper, we propose a streamlined technique for learning token level language ID, which we term *Concatenated Tokenizer*. Unlike previous approaches of "aggregating" tokenizers that take the union of the per-language tokenizer vocabularies (Li et al., 2021) or create a shared sub-word token set across languages (Pratap et al., 2020a), in the concatenated tokenizer method we reuse monolingual tokenizers and map them to mutually ex-

clusive label spaces for each language. This helps provide explicit language information to the ASR model while training and leads to inexpensive prediction of token level LID at decoding time.

The main contributions of the paper are as follows:

1. A scalable and extensible synthetic code-switching ASR data generation pipeline that allows us to generate a corpus of any size, online (e.g. during training) or offline, from strictly monolingual data sources.

2. The Concatenated Tokenizer method which can effectively utilize pre-existing monolingual tokenizers and provide token level LID information while learning multilingual and CS ASR models.

3. We demonstrate CS speech recognition capabilities of the proposed unified ASR model on real world data for two language pairs and spoken language identification capabilities on the out of distribution FLEURS evaluation dataset.

## 2 Multilingual and Code-Switching ASR

Modern Natural Language Processing (NLP) and ASR models use tokenizers to represent text (Kudo and Richardson, 2018). The traditional approach requires that a new tokenizer is learned for each language and domain. In ASR, this tokenizer is also used to reduce the target sequence length to satisfy CTC requirements under aggressive downsampling with respect to original audio length (Graves et al., 2006). In this section, we discuss the proposed concatenated tokenizer and the synthetic code-switching data generation pipeline.

### 2.1 Concatenated Tokenizers

When training multilingual ASR models, monolingual training sets typically have significantly different characteristics (e.g. total size, quality, noise levels, etc.), requiring experimentation of combining them with different ratios for an optimal outcome. Training a different tokenizer on the combined mixture of datasets for each experiment becomes a logistical challenge, while the resulting model must always use the exact same tokenizer with which they it was trained. Synthetic code-switching training datasets present an additional challenge since the tokenizer learns the purely synthetic co-occurrence of adjacent tokens from dif-

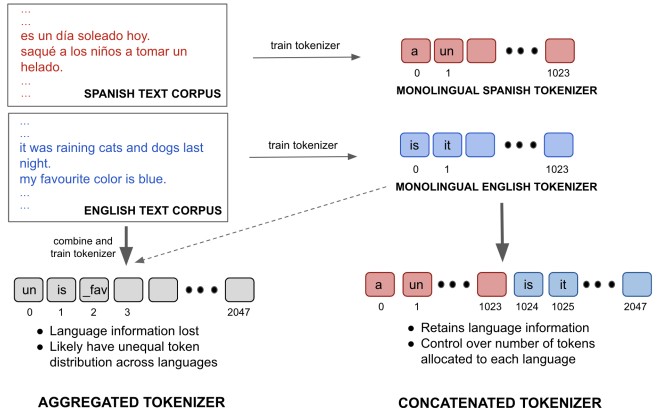

Figure 1: Aggregated vs Concatenated (proposed) tokenization approaches for a bilingual English-Spanish example. Spanish text and tokenizer is represented in red, English text and tokenizer in blue.

ferent languages, which is unlikely to occur in real code-switching data. Finally, when training a single tokenizer on multilingual data, we must disregard the LID information for each token and need to rely on an external technique if retention of LID is desirable.

We propose the concatenated tokenizer technique to mitigate the above issues. Fig. 1 illustrates this approach when training a bilingual English-Spanish model with the vocabulary size of 2K. In the traditional approach, text transcripts are mixed in some proportion and a tokenizer is trained on the joint text corpus. LID information is lost and would need to be re-supplied if desired. For a CS use case, training a tokenizer on a synthetic code-switching dataset directly results in it learning arbitrary transitions between language samples, and is likely to be avoided.

In the concatenated tokenizer method, we train English and Spanish tokenizers with a 1K vocabulary size each on the corresponding monolingual datasets separately. We allocate the range of IDs from 0 through 1023 to English and 1024 through 2047 to Spanish. To achieve that, after tokenization, we shift each Spanish token by 1024 to ensure that it lands in its allocated range. The concatenated tokenizer has, therefore, also 2K tokens. During training, we use the English tokenizer (shown in blue) to tokenize each English sample segment, and the Spanish tokenizer (red) to tokenize each Spanish segment. At inference time, the model predicts a sequence of token ids. If the token ID is in the range from 0 to 1023, we know that it is an English token, and we use the English tokenizer to convert

it to text. Similarly, tokens in the range from 1024 to 2047 are Spanish and are sent to the Spanish tokenizer for detokenization. Language ID information is embedded in the ID of each token and can be used in downstream processing of the resulting text segments. We name our method concatenated tokenizer because such tokenizer effectively contains more than one separate monolingual tokenizer with its preserved non-overlapping token space. In the above example, we chose to allocate the same number of tokens to English and Spanish, but that certainly does not need to be the case when dataset sizes are very different.

Note that the concatenated tokenizer method differs significantly from the standard technique of training an aggregated tokenizer on a mixture of transcripts and then re-injecting the language information into the tokenized sequences via special LID tokens (Seki et al., 2019), (Radford et al., 2022). In the latter approach, the special LID tokens indicate only the beginning and end of each monolingual span of text.

The design of the concatenated tokenizer allows us to easily suppress specific languages from inference when it is known that the audio does not contain them. We simply do not need to compute probabilities for token IDs in the ranges corresponding to the suppressed languages, simultaneously improving performance. Fig. 2 illustrates how this works with the CTC decoder. Conversely, when adding language(s) to the decoder, we can transfer existing token weights via weight surgery, initializing weights for the incremental language(s) from scratch. The same idea works with the Transducer decoder as well. This allows for a better decoder initialization while training multilingual model from monolingual checkpoints, while improving convergence time.

## 2.2 Synthetic data generation for Code Switching ASR

Synthetic code-switching data generation was an essential step in our work. It enabled us to effectively use the monolingual training data available at our disposal to generate a diverse set of CS speech training samples which then was utilized by our model for training. We had to be careful in the data generation strategy to ensure that we didn't introduce a bias of any kind that would make model training easier but would lead to poor performance on real world code-switching data. For example,

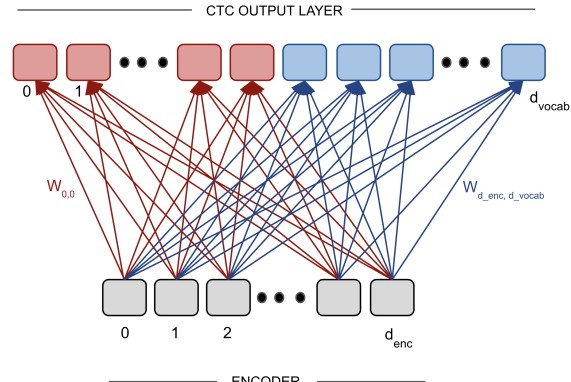

Figure 2: Diagram illustrating the benefits of concatenated tokenizers for easy addition/suppression of languages in multilingual ASR models. For simplicity, we show a single output step of a bilingual ASR model with a CTC decoder consisting of one feed-forward fully connected layer (FC) with weights $W$ that maps encoder representation (dimension $d_{enc}$) to token logits (dimension $d_{vocab}$, blank symbol omitted). The concatenated tokenizer has two languages marked by red and blue. Due to the non-overlapping token mappings for different languages in the concatenated tokenizer, the FC weights can easily be separated and modified independently.

if we simply stitched different speech samples together it would cause inconsistencies and the different amplitudes and background conditions can give the model easy clues for learning this generated data. Such inconsistencies would not be found in real life examples, and hence the model would not be able to generalize its performance. Furthermore, we didn't want to bias the generated samples to start from or end with a particular language, e.g. English.

We used the algorithm detailed in Fig. 3 for generating synthetic CS speech data for two or more languages from their monolingual speech corpus. Each language was assigned a sampling frequency. For each synthetic CS sample we define a max and min duration, which are controllable parameters. This allows us to generate samples with a specific duration distribution and also ensures that the samples are of similar lengths which leads to lesser padding during batching, leading to more effective utilization of the data. To have a control over leading, trailing silences in the synthetic sample as well as the gap between concatenated samples, we introduce three parameters: duration beginning silence, duration ending silence, duration joining silence. In the current implementation we use silence, but this can easily be extended to adding noise with a desired SNR. The next step in the al-

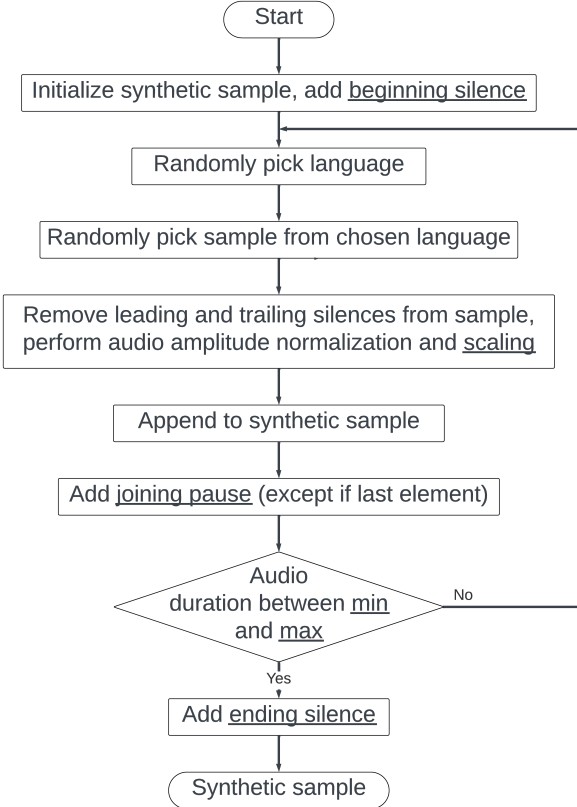

Figure 3: Flowchart of the synthetic CS sample generation process for two languages. The controllable hyperparameters have been underlined. The process can be used for both online synthetic data generation in the dataloader or offline creation of synthetic speech corpus as discussed in Section 2.2.

gorithm is removal of leading and trailing silences. This ensures that we extract only the speech portion from the individual utterances and discard any silences in the beginning or end of the utterance. This allows us to have complete control over the leading, joining, and trailing silences in the generated sample using our tunable duration parameters explained earlier. In our current implementation we use an amplitude based threshold for removing silence, but this would be extended to voice activity detection (VAD) in the future iterations. Another important step in our algorithm is audio amplitude normalization and scaling. We perform peak amplitude normalization for each sample before concatenation and multiply the normalized sample with the controllable scaling parameter to ensure that all samples are in a similar amplitude range before joining, removing the amplitude bias from individual datasets that provide the monolingual samples. It should be noted that a similar synthetic speech data generation idea was proposed in (Seki et al.,

2018), but our approach is more customizable and general due to the larger number of controllable parameters. We found this technique to be useful in generating synthetic samples that are longer and similar in duration, which accelerates training. Synthetic CS data generation approaches have also been explored in the text domain, for training multilingual language models (Winata et al., 2019) and translation models (Gupta et al., 2020; Tarunesh et al., 2021).

In our implementation, we provide both an offline and online version of the synthetic data generation pipeline. In the offline version, the generated synthetic corpus using the proposed algorithm is stored explicitly and can be used to train the ASR model. In the online version, the synthetic sample generation process happens in the dataloader, and is used to feed samples to ASR model for training. The online data generation approach provides the advantage of not having to save the generated synthetic corpus, and hence can be used to rapidly experiment with different language ratios and other parameter permutations, generating massive synthetic training CS ASR corpora with no disk space overhead. The code for the data generation process is open-sourced and available in NeMo toolkit[1].

## 2.3 Spoken Language Identification

Spoken language identification refers to the task of identifying the language of a given utterance directly from audio (Li et al., 2006). This task is critical for CS ASR because it enables us to reuse monolingual models to re-score CS decoded output if we can predict which language was spoken when. The proposed concatenated tokenizers fit in here perfectly as they also contain the information of the language that each predicted token belongs to. To calculate the efficacy of concatenated tokenizer for utterance level spoken language identification, we take the maximum over the predicted language for each token in the sentence. To ensure a fair comparison, we trained our models with the datasets described in Section 3.1 but evaluated spoken language identification performance on the blind test sets of the FLEURS [26] dataset.

---

[1]https://github.com/NVIDIA/NeMo/scripts/speech_recognition/code_switching

## 3 Experimental Setup

### 3.1 Datasets

We used LibriSpeech (Panayotov et al., 2015)($\sim$ 960 hours) as the English corpus. For Spanish, we compiled a dataset ($\sim$ 1300 hrs after basic cleaning) consisting of Mozilla Common Voice 7.0 (Ardila et al., 2020), Multilingual LibriSpeech (Pratap et al., 2020b), Voxpopuli (et al, 2021) and Fisher (Graff et al., 2010) (all Spanish). For Hindi training we used the ULCA dataset (Dhuriya et al., 2022) ($\sim$ 2,250 hrs after basic cleaning).

For English-Spanish (en-es) and English-Hindi (en-hi) synthetic CS data generation we follow the approach outlined in Section 2.2 to generate a 10K hour training corpus with the following parameters: max sample duration 19 sec, min sample duration 17 sec, silence duration 0.02 sec, ending silence duration 0.02 sec, joining silence duration 0.1 sec. Using the same parameters, we generated 10 hour synthetic bilingual CS test sets using monolingual test sets for both language pairs. Language sampling probabilities were a parameter; we experimented with multiple ratios in the course of our experiments.

We chose the Miami Bangor corpus (Deuchar et al., 2014), which consists of full conversations, as the Spanish out-of-distribution CS test set. Individual interactions were extracted using provided timestamps. All utterances less than 2 seconds were removed. The final evaluation set has 16 hours with 16620 utterances and 35 unique characters. As the Hindi CS test set, we use the MUCS 2021 corpus (Diwan et al., 2021). We performed basic cleaning, leading to a 5 hour set with 3136 samples and 89 unique characters.

### 3.2 Models and Experiments

We used the Conformer-RNNT Large model (Gulati et al., 2020) ($\sim$ 120 M parameters, no external LM) and trained it for 200 epochs using the AdamW optimizer and Noam scheduler with a 20k steps warmup, 0.0015 peak learning rate and $10^{-6}$ minimum learning rate. We performed the following experiments:

- **Monolingual**: We trained monolingual English, Spanish, and Hindi ASR models (see Section 3.1). For each language, we trained an SPE unigram tokenizer (Sennrich et al., 2016) with a vocabulary size of 1024.

- **Bilingual**: We trained bilingual English-Spanish and English-Hindi models. We mixed the monolingual datasets in different ratios with the general idea of over-representing the smaller dataset. We trained two classes of models, one with the concatenated tokenizer (Section 2.1) and another with a regular (aggregate) tokenizer trained on the combined text corpus in the 1:1 ratio. We further performed an initialization study for both language pairs by training from scratch or starting from either monolingual checkpoint.

- **Code-Switching (CS)**: We trained CS English-Spanish and English-Hindi models using the synthetic code-switching data highlighted in Section 3.1. We experimented with training from scratch and also the corresponding bilingual (non-CS) model. Further, we investigated using both concatenated and aggregate tokenizers in all the scenarios.

- **Language Identification**: We used the English-Spanish and English-Hindi concatenated tokenizer trained during the bilingual CS experiments to perform utterance level language identification on the English (en_us_test, 647 samples), Spanish (es_419_test, 908 samples), and Hindi (hi_in_test, 418 samples) speech samples from the FLEURS set (Conneau et al., 2023).

## 4 Results and Discussion

In this section, we present results for the experiments outlined in Section 3.2. Table 1a shows performance of monolingual, bilingual and CS English-Spanish models with different tokenizers on the English Librispeech and Spanish Fisher test sets. We used dataset mix ratio (English to Spanish) of 2:1 for training the bilingual model in order to balance the training set. The Fisher test set was chosen to represent model performance on Spanish because it was the hardest (highest WER) out of the four Spanish datasets mentioned in Section 3.1. Similarly, Table 1b presents the results for the different models for English-Hindi language pair. We used dataset mix ratio (English to Hindi) of 2:1 as well for the bilingual model, again aiming to balance the training set. Results were averaged across three runs and averaged (Liu et al., 2018) over the five best model checkpoints.

Table 1: Monolingual evaluation set results for the English-Spanish and English-Hindi models. We present WER(%) (lower is better) for multilingual (ml) and code-switched (cs) models trained with concatenated (con) and aggregate (agg) tokenizers vs monolingual baselines. We observe that the use of the concatenated tokenizer does not hurt model performance while adding the ability to predict LID for each token.

(a) English-Spanish results on the monolingual English Librispeech test-other and Spanish Fisher test sets.

| Model | Tokenizer | English LS test-other | Spanish Fisher-test |
|-------|-----------|-----------------------|---------------------|
| en | mono | 5.29 | 98.37 |
| es | mono | 85.68 | 16.14 |
| ml | agg | 5.00 | 16.37 |
| ml | con | 5.14 | 16.72 |
| cs | agg | 5.38 | 16.35 |
| cs | con | 5.28 | 16.42 |

(b) English-Hindi results on the monolingual English Librispeech test-other and Hindi ULCA eval sets.

| Model | Tokenizer | English LS test-other | Hindi ULCA |
|-------|-----------|-----------------------|------------|
| en | mono | 5.29 | 100 |
| hi | mono | 100 | 10.53 |
| ml | agg | 5.00 | 10.78 |
| ml | con | 5.14 | 10.73 |
| cs | agg | 5.42 | 11.35 |
| cs | con | 5.29 | 11.64 |

Table 2: Performance comparison of the code-switched (cs) English-Spanish and English-Hindi models trained with concatenated (con) and aggregate (agg) tokenizers on both synthetic and real world blind CS evaluation datasets. The performance of the multilingual (ml) models has also been reported as a benchmark. We observe that cs models significantly outperform ml models, highlighting the advantage of using the proposed synthetic CS data for training.

(a) Code-switched English-Spanish models: WER(%) on synthetic and Miami-Bangor CS evaluation sets.

| Model | Tokenizer | synth | Miami |
|-------|-----------|-------|-------|
| cs | agg | 5.51 | 50.0 |
| cs | con | 5.50 | 53.3 |
| ml | agg | 16.52 | 58.78 |
| ml | con | 24.08 | 63.54 |

(b) Code-switched English-Hindi models: WER(%) on synthetic and MUCS CS evaluation sets.

| Model | Tokenizer | synth | MUCS |
|-------|-----------|-------|------|
| cs | agg | 6.55 | 30.3 |
| cs | con | 6.57 | 28.78 |
| ml | agg | 35.70 | 62.18 |
| ml | con | 53.01 | 100 |

The results for English-Spanish CS experiments are highlighted in Table 2a and for English-Hindi in Table 2b. The numbers for monolingual models are not reported on the code-switching evaluation datasets as they are relatively poor, as expected. When multilingual models are used to decode code-switched speech, we observe that they tend to stick with the language in which the utterance started and are not able to switch between languages as they occur within the utterance. This is evidenced by the correspondingly high WERs in Tables 2a and 2b, and is consistent with the fact that the models did not encounter code-switched data during training.

To illustrate, here are sample transcripts from one English-Spanish code-switched audio:

**ML model output**: con qué departamento puedo dejar fitbax sobre mi experiencia de compra en la tienda que estaba ubicada en one tú threforme ave

**CS model output**: con qué departamento puedo dejar feedbacks sobre mi experiencia de compra en la tienda que estaba ubicada en one two three fourth avenue

We can see that the ML model is not able to switch from Spanish (majority language) to English (underlined for easier visual comparison). On the other hand, the output of the CS model is 100% accurate.

Finally, the results of the Language Identification experiments for all the three languages are presented in Table 3. In the following, we dive deeper into the results and discuss the findings.

## 4.1 Bilingual models, effect of model initialization and tokenizers

From Tables 1a and 1b we observe that the bilingual and CS models achieve comparable performance to monolingual models on respective monolingual evaluation sets. This was seen for both the language pairs considered: English-Hindi and English-Spanish. It is an interesting result as this allows us to use a single bilingual code-switched model instead of creating two separate monolingual models for each language. Initializing training from either monolingual checkpoint, while accelerating training, did not improve the final WER

Table 3: Spoken language identification using English-Spanish and English-Hindi concatenated tokenizers on the FLEURS dataset.

| Language | # of samples | LID accuracy |
|---|---|---|
| English (en_us_test) | 647 | 98% (632/647) |
| Spanish (es_419_test) | 908 | 100% (908/908) |
| Hindi (hi_in_test) | 418 | 99% (414/418) |

for both language pairs considered. Using either a concatenated or an aggregate tokenizer led to similar performance. However, the concatenated tokenizer provides additional benefits, such as language identification and multilingual LM rescoring, as discussed in the Section 4.3.

### 4.2 Code-Switching models, effect of model initialization and tokenizers

Table 1a presents the performance of the English-Spanish CS ASR models on monolingual test sets: Librispeech and Fisher. Table 2a presents the corresponding results on the code-switched sets: synthetic and the Miami Bangor corpus. Similarly, for the English-Hindi code-switched ASR models, Table 1b presents the performance on monolingual test sets: Librispeech and ULCA, while Table 2b summarizes the results on the code-switching test sets: synthetic and MUCS. For both language pairs, we observed that initializing the code-switched model from the multilingual checkpoint leads to better results and faster convergence as opposed to initializing the model from scratch or from either monolingual checkpoint. We also experimented with different language dataset mix ratios and determined that the best results are achieved when the code-switched dataset is roughly balanced. This may require oversampling of the smaller language.

In (Weller et al., 2022), the authors reported a performance of 53% on the Miami Bangor corpus, which shows that our code-switched models perform competitively with the state-of-the-art on real world samples, while being trained purely from synthetic code-switched data. Another important observation is that the concatenated tokenizer performs just as well as the aggregate tokenizer for the CS models and therefore should be preferred given the additional benefits that it provides. Concluding

the discussion, we now have a single model that performs well on monolingual, bilingual, as well as code-switched data.

### 4.3 Concatenated Tokenizers and Language identification

Table 3 presents utterance-level spoken language identification performance of the English-Spanish and English-Hindi models trained with the concatenated tokenizer on the test sets of the FLEURS dataset. We observe that these models are very accurate at predicting the language of the utterances directly from speech samples. We find it to be significant, since these samples are out of distribution and were not seen by the model during training.

## 5 Conclusion

In this paper we investigate training of bilingual and code-switching models using purely monolingual datasets. We propose two novel techniques: (1) a real-time and offline synthetic code-switching data generation pipeline and (2) the concatenated tokenizer method, which allows the model to predict language ID directly at the level of individual tokens. We use these two techniques to train CS ASR models and find that they match monolingual model performance on monolingual evaluation benchmarks while performing significantly better on code-switching data. We evaluate model performance against synthetic CS test sets as well as real world English-Spanish Miami Bangor and English-Hindi MUCS corpora. In addition, we find that the models display strong performance on LID detection, which we measure using the FLEURS dataset. Performance of models trained with the novel concatenated tokenizer is similar to models trained with the regular aggregate tokenizer, while offering the additional benefit of LID detection. The results suggest that these approaches could be extended to additional languages without increasing model architecture complexity. Further, the excellent LID capabilities of concatenated tokenizer models can enable us to use monolingual language models to rescore and further improve code-switched model predictions. All of this has implications for further research. The code and model weights have been released publicly in NeMo[2].

---

[2] https://github.com/NVIDIA/NeMo

## Limitations

In this work we present techniques that enable the development of code-switching speech recognition models exclusively from monolingual data sources. To validate the efficacy of the proposed work, we experimented with two language pairs: English-Spanish (en-es) and English-Hindi (en-hi). We selected en-es due to the prevalent bilingual nature of English and Spanish, while en-hi was chosen as Hindi and English possess distinct character sets, thereby allowing us to assess the robustness of our approach. However, we would need to perform experiments with a more diverse set of language pairs to validate if the methods work in general. Furthermore, more experiments are warranted to see if the concatenated tokenizer expands to more than two languages used at a time. As the concatenated tokenizer assigns mutually exclusive token spaces for each language, its size increases with the inclusion of additional languages. This scalability challenge may potentially impede the construction of massive multilingual models. By addressing these limitations through future research endeavors, we can enhance the comprehensiveness and applicability of our findings in the realm of code-switching speech recognition.

## Ethics Statement

We adhere to and endorse the principles outlined in the ACL Ethics Policy. Our work on synthetic code-switched ASR holds the potential to offer far-reaching benefits across a spectrum of languages, spanning from widely spoken to less common ones. By alleviating the challenges associated with data collection, our research contributes to the advancement of a more diverse and equitable linguistic landscape. Furthermore, our exploration of multilingual models not only streamlines the computational demands of training and deployment but also fosters resource efficiency by consolidating the utility of numerous monolingual models. Finally, we affirm our commitment to transparency and openness by sharing all code and models used in this study, which were exclusively trained on publicly available datasets, and making them accessible online.

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
