# OpenReview forum: "Unified Model for Code-Switching Speech Recognition and Language Identification Based on Concatenated Tokenizer"
_EMNLP/2023/Workshop/CALCS — EMNLP 2023 Workshop CALCS_

### Official Review · Reviewer_wz2f · 2023-09-20
**Review of CCS and LID paper**

**Rating:** 4
**Confidence:** 5

**Review:**

Summary: The paper used existing monolingual dataset to create code-switching corpus and used improved language identification rule to recognize code-switching.
Appropriateness: The paper's topic definitely fits for the workshop.
Impact of the task: The paper's approach is new especially the way of creating new code-switch corpus.
Impact of methods: The methodology logic is good but lack of showing examples.
Results: the results only show the LID accuracy but does the model only use LID evaluation?
Disadvantages:
Context: The paper needs to improve English because it is hard to follow up by some concepts. I suggest authors used more examples to explain how to create CS from monolingual instead of large texting explanation. The examples can be how to use those rules to mix or get LID from one sentence. The context of each sentence of the paper needs more connection words or illustrate the idea more clearly from simple to deeper.
Dataset: The paper should show some examples of creating CS dataset. Does the CS dataset follow the grammar since CS speaker has different habits for CS grammar? How you keep the sentence to make sense and have appropriate grammar?

**Candidate For Best Paper:**

No

**Reason For Best Paper:**

N/A

**Related:**

5: It is very related to the workshop.

---

### Official Review · Reviewer_VwGu · 2023-10-02
**A solid experimental study of data generation for multilingual ASR in the presence of CS data**

**Rating:** 3
**Confidence:** 3

**Review:**

This paper studies ways to develop ASR systems that would be robust to code-switched inputs based only on monolingual data. For this the author develop a methodology for generating artificial training data combining in an acoustically controlled fashion samples from two monolingual corpora, providing the authors with the training resources for a bilingual ASR decoder. Another innovation of the paper is to first independently train and then merge subword lexica for the two languages under consideration, so that each output token is associated unambiguously with one language. With this trick, performing LID on the decoding subword sequences becomes entirely trivial.

Experiments are performed with two pairs of languages (en-es - spanglish; en-hi, hinglish) on multiple test sets containing either monolingual data, or code-switched data. Trained on artificial data, their system is able to decode two monolingual datasets with the same performance as a monolingual ASR, but is claimed to be way more robust when it comes to process CS inputs (no comparison is provided here).[*] Overall, the system seems to be on par with a bilingual system trained with a mixture of data from two languages. The use of a "concatenated tokenizer" (more precisely disjoint "subword inventories") does not seem to hurt (nor improve) ASR performance; it is of great help for LID.

This is a serious and overall clear and well written paper. The methodological contributions are small, but the evaluation on two language pairs seems solid and the claims are well supported. Results on an actual difficult dataset show that the problem is far from being solved - in particular training a speech or language model to spontaneously switch as a human would do remains a complicated issue.

Using mixed-language data within utterances has been performed repeatedly to improve multilingual LMs or translation models - this should be reflected in the reference list. Note that in these references (as in the current paper) I would object to calling the resulting articifial corpus "code-switched corpus" -- as it lacks the linguistics traits of CS languages, it would more properly be refered to as "code-mixed corpus".

Missing refs:
- https://arxiv.org/abs/1909.08582
- https://aclanthology.org/2020.findings-emnlp.206/
- https://arxiv.org/abs/2107.06483
- https://arxiv.org/abs/2105.04846

[*]: One result that is missing in Table 2 is the performance of the bilingual system. The authors say that the monolingual system is very bad - how about the multilingual system ?

**Candidate For Best Paper:**

No

**Reason For Best Paper:**

No recommendation, no reason given

**Related:**

5: It is very related to the workshop.

---

### Official Review · Reviewer_Upfz · 2023-10-05
**Unified Model for Code-Switching Speech Recognition and Language Identification Based on Concatenated Tokenizer**

**Rating:** 4
**Confidence:** 5

**Review:**

This paper introduces a simple but effective and scalable method for creating synthetic CS ASR data. The explored data pairs are Hindi-English and Spanish-English. The experimental setup is well done, and the paper is well written. Also, the empirical results are strong. Therefore,  I can not find any risk in accepting this paper to the workshop. Of course, I would love to use a broader language pair set to evaluate this method on, but I also recognize that there are not many evaluation benchmarks for ASR with code-switching.

**Candidate For Best Paper:**

No

**Reason For Best Paper:**

-

**Related:**

5: It is very related to the workshop.